# Improved Re-Parameterized Convolution for Wildlife Detection in Neighboring Regions of Southwest China

**DOI:** 10.3390/ani14081152

**Published:** 2024-04-10

**Authors:** Wenjie Mao, Gang Li, Xiaowei Li

**Affiliations:** School of Mathematics and Computer Science, Dali University, Dali 671003, China; moonjay1013@163.com (W.M.); lixiaowei_xidian@163.com (X.L.)

**Keywords:** reparameterization, object detection, wildlife technology, monitoring wildlife, deep learning

## Abstract

**Simple Summary:**

In response to the quantitative demand for the practical deployment of applications and the need to enhance the detection accuracy of wildlife in complex field environments in southwest China and neighboring regions, we have refined the wildlife detection algorithm based on re-parameterized convolution. This refinement is specifically targeted at addressing the challenges posed by the low quality of wildlife images captured by camera traps and the limitations of traditional object detection algorithms in feature extraction capability. To address these issues, we have introduced a series of improvement schemes. As a result of these enhancements, there has been a noteworthy improvement in both the accuracy of wildlife detection and the speed of model inference. This advancement offers a convenient and efficient method for the preliminary detection in the context of automated wildlife monitoring.

**Abstract:**

To autonomously detect wildlife images captured by camera traps on a platform with limited resources and address challenges such as filtering out photos without optimal objects, as well as classifying and localizing species in photos with objects, we introduce a specialized wildlife object detector tailored for camera traps. This detector is developed using a dataset acquired by the Saola Working Group (SWG) through camera traps deployed in Vietnam and Laos. Utilizing the YOLOv6-N object detection algorithm as its foundation, the detector is enhanced by a tailored optimizer for improved model performance. We deliberately introduce asymmetric convolutional branches to enhance the feature characterization capability of the Backbone network. Additionally, we streamline the Neck and use CIoU loss to improve detection performance. For quantitative deployment, we refine the RepOptimizer to train a pure VGG-style network. Experimental results demonstrate that our proposed method empowers the model to achieve an 88.3% detection accuracy on the wildlife dataset in this paper. This accuracy is 3.1% higher than YOLOv6-N, and surpasses YOLOv7-T and YOLOv8-N by 5.5% and 2.8%, respectively. The model consistently maintains its detection performance even after quantization to the INT8 precision, achieving an inference speed of only 6.15 ms for a single image on the NVIDIA Jetson Xavier NX device. The improvements we introduce excel in tasks related to wildlife image recognition and object localization captured by camera traps, providing practical solutions to enhance wildlife monitoring and facilitate efficient data acquisition. Our current work represents a significant stride toward a fully automated animal observation system in real-time in-field applications.

## 1. Introduction

Biological resources are the natural foundation for human survival and development, and wild animals are an important component of biological resources. Before the beginning of the 21st century, the value of wild animals was generally ignored or underestimated by the international community [1]. Therefore, wildlife conservation faced various threats, and the commercial trade of wild animals gradually led to endangered species in southwestern China and surrounding countries [2]. Protecting wildlife is of great significance for sustainable development and is a key concern for China and other countries in the world. Scientifically and effectively monitoring wild animals leads to the acquisition of their information, including images, geographic distribution, and environmental conditions. By analyzing this information, we can understand the species categories, quantities, and habitat conditions of wild animals, thereby aiding decision-making for biodiversity conservation. In addition, the role of small mammals as carriers or reservoirs of zoonotic diseases in urban forest areas [3] is significant; therefore, wildlife is monitored to prevent the spread of disease and possible problems associated with contact with wildlife and humans, which is important for the One Health approach [4]. However, the widespread distribution, low population density, unpredictable behavior patterns, and sensitivity to interference of wildlife pose significant challenges to monitoring work. Traditional wildlife investigation techniques mainly include manual investigation, collar tracking, trap lines, and acoustic tracking using sound recording instruments [5,6,7]. These traditional techniques are applicable to all animal species and require certain installation skills. Except for acoustic tracking, all other methods inevitably interfere with the normal activities of wildlife. However, acoustic tracking is limited due to the need for wildlife to make sounds, and it is unable to capture morphological information about the animals. In contrast, round-the-clock monitoring can be carried out by widely deploying infrared cameras in the areas where wild animals are found.

The study of wildlife distribution and behavioral ecology is one of the keys to the targeted protection of wildlife, non-invasive monitoring is the basic requirement for wildlife monitoring, and digital imaging technology is one of the important ways to realize this. Camera traps are widely used in the field of wildlife conservation. The well-known Snapshot Serengeti (SS) project [8] acquired image data by this method. Although infrared cameras capture a vast number of images, the proportion of useful ones is typically minimal. Most of these images are “empty” photographs triggered falsely by infrared cameras, devoid of any animal objects, necessitating manual coarse screening for filtering. Moreover, the images acquired by infrared cameras often encounter challenges such as overexposure, blurring, small object sizes, and other issues resulting in poor image quality, all attributable to the variability and intricacy of the field environment. Given the complexity inherent in these images, their raw forms pose difficulties in analysis, necessitating the extraction of relevant information through visualization techniques. For a considerable period, human labor served as the primary method for identifying and tallying animals depicted in images. It still takes years for experts and citizen scientists to manually annotate millions of images in SS programs [9].

To avoid these problems, various automatic and semi-automatic detection algorithms for wildlife have been employed, especially for images from UASs [10,11,12]. However, these methods are not suitable for detecting objects with similar gray values to complex backgrounds [13]. In order to detect objects in complex environments, various machine learning methods are applied to object localization with satisfactory results in relatively simple scenes [14], but they are not sufficient to detect complex animal features such as structure, texture, and morphology [15]. Recently, multilayer neural network-based deep learning methods have made impressive breakthroughs in pattern recognition in several fields such as wildlife image classification [16,17], facial recognition [18], and object detection [19,20]. As a result, wildlife detection is rapidly transforming into a data-rich discipline driven by advances in computer vision technology, and it has been applied to automate the detection of a wide range of wildlife species.

Object detection represents a fundamental task in computer vision, which involves detecting and localizing one or more classes of objects in image data. The most prevalent convolutional neural network (CNN)-based object detectors include SSD (Single Shot MultiBox Detector) [21], the RCNN series [22,23,24], EfficientDet [25], and the YOLO series [26,27,28]. Despite showing excellent performance in detecting wildlife species, the current state-of-the-art deep learning algorithms lead to missed and false detections of endangered species [29] with unique body textures, shapes, sizes, and colors due to their insufficient fine-grained feature extraction capabilities. Accurate detection and localization tasks are challenging due to species’ biological habits that may result in significant variations in lighting conditions, low visibility, high closeness, overlap, the coexistence of multi-target classes with a variety of aspect ratios, and other morphological features [30]. Wildlife visual similarities, complex backgrounds, difficulty in distinguishing between species and their surroundings, and various other critical factors provide additional challenges and difficulties for state-of-the-art wildlife detection models [31].

To solve the problem of difficult and extremely inefficient labeling after manual filtering, deep learning algorithms need to be utilized. One feasible strategy involves deploying models on embedded devices to collaborate with camera traps, facilitating the automated detection of wildlife objects. However, common embedded devices are constrained by factors such as power consumption, SoC size, and manufacturing costs, resulting in limited image processing capabilities. Moreover, considerations must be made regarding hardware support for deep learning frameworks, necessitating professional assistance in the form of dedicated deep learning devices like the NVIDIA Jetson. However, given the platform’s limited computational power, deployment necessitates a lightweight model while ensuring high robustness across diverse situations. Scientists [32] deployed high-definition cameras in the field to automatically capture wildlife images and collect network images to form the Yunnan Tongbiguan dataset, used the lightweight network structure MobileNet instead of the YOLOv4 backbone, then introduced migration learning to solve the problems of insufficient training data and network fitting difficulties, which improved the average accuracy of the model, but the detection efficiency was not high. Aiming at the problems of a high detection error and leakage rate caused by the low contrast between the background and the target of the forest animal images captured by the camera traps, strict occlusion, and overlap, as well as the data imbalance, other scientists [19] used a Swin Transformer and self-attention mechanism to improve the perceptual field for feature extraction based on the YOLOv5s model, by modifying the loss function to reduce false detections and speed up the convergence of the model; although the enhancement effect is obvious on the self-constructed forest wildlife dataset, the same does not pay attention to the inference cost of the model’s practical application, and it is not friendly for deploying the application to work with the camera trap. Compared to their work, WilDect-YOLO [20] considered that the morphology of endangered wildlife has high variability and a complex background, and after introducing strategies such as residual block, Spatial Pyramid Pooing (SPP), and Path Aggregation Network (PANet) in YOLOv4, it improved to some extent the detection accuracy and achieved real-time detection requirements; however, most of the self-constructed wildlife datasets are from web images, which are quite different from the images taken by camera traps.

The primary goal of this study is to realize efficient automatic wildlife monitoring, high accuracy, and fast detection, and for the recognition of monitoring images captured by camera traps. We propose a wildlife detection algorithm based on improved YOLOv6 [33] to cope with complex field environments. First, to improve the detection accuracy of the detector, taking the re-parameterized convolution as the research object, we analyze the focus of the detector on the feature map at different stages, use asymmetric convolution to enhance the feature extraction ability of the Backbone network’s convolutional block, and improve the detector performance with merely an increase in training costs. Second, after pruning the feature fusion network, we use the architectural design of the dual detection head to improve the inference speed of the model. Then, the box regression loss function CIoU loss is introduced to enhance the detector accuracy and accelerate the network convergence. Finally, due to the problem of serious degradation of detection accuracy after INT 8 precision quantization using re-parameterized convolution, RepACID-Optimizer, which is more suitable for the model in this paper, is improved for asymmetric convolutional branching based on RepOptimizer [34], and a pure VGG-style [35] single-branching model is trained using the improved optimizer to ensure the stability of the accuracy of the quantization of the detector. The final model is deployed to the NVIDIA Jetson Xavier NX device with higher detection performance than the latest YOLO algorithms [33,36].

## 2. Materials and Methods

### 2.1. Data Acquisition and Pre-Processing

#### 2.1.1. SWG Dataset

The images in the wildlife Saola Working Group (SWG) Camera Traps 2018–2020 dataset [37] were collected from a sequence of 436,617 camera trap images from 982 sites in both Vietnam and Laos, with a total of 2,039,657 images, of which about 12.98% of the images were labeled as empty. Data were downloaded from the LILA BC website [38], showing their common names in regular English and stating the genus and species to which the wildlife belongs in italicized Latin scientific names. The tags were divided into 120 categories, mainly at the species level, and the tags with the highest number of images were “Eurasian Wild Pig (*Sus scrofa*)”, “Large-antlered Muntjac (*Muntiacus vuquangensis*)”, and “unidentified Murid (*Muridae*)”.

The Saola Working Group (SWG) aims to save the critically endangered species called the Saola (*Pseudoryx nghetinhensis*) by carrying out its activities within the range of the Saola in the Lao People’s Democratic Republic and Vietnam, with the main sites shown on the maps below. As the SWG are unable to work in all areas where the Saola are likely to be found, there is a need to identify a small number of priority sites in Lao PDR and Vietnam, where the Saola are most likely to be found. Lao PDR includes Bolikhamxay, the Khoun Xe Nongma Protected Area (KXNM), Xe Kong province, and Nakai-Nam Theun (NNT). Vietnam includes Pu Mat National Park (NP), Quang Tri-Quang Binh, and Vu Quang. By reading the SWG’s annual reports, we obtained information on the areas where camera traps are deployed, and the number of them, as shown in Figure 1 and Table 1.

To validate the performance and robustness of the algorithm we improved, the publicly available dataset PASCAL VOC [39] in the field of object detection was selected for this study, which contains 20 categories common to daily life as a universal evaluation criterion for detector performance. A subset of the wildlife SWG dataset was selected, specifically using the 7 wildlife species in the SWG project that have been manually labeled with detection frames and have image counts between 100 and 500 (hereafter referred to as SWG7), namely the following (as shown in Figure 2a–g): Spotted Linsang (*Prionodon pardicolor*), Blue Whistling Thrush (*Myophonus caeruleus*), Red Muntjac (*Muntiacus muntjak*), Chevrotain (*Tragulus kanchil*), Large Indian Civet (*Viverra zibetha*), Macaque (*Macaca mulatta*), and Red Shanked Douc (*Pygathrix nemaeus*). SWG7 was selected for model training because the number of images of these 7 species captured by the infrared camera was small, the objects were biologically distinct from small to large, and the background information contained complex environmental conditions such as nighttime darkness, woodland, and scrub, which could reproduce most of the natural impacts encountered by the camera trap deployment in the field. In addition, *P. nemaeus* is an endangered species, *V. zibetha* is listed in China’s national first-level protected animal list, *M. mulatta* and *P. pardicolor* are listed in China’s national second-level protected animal list, *M. caeruleus* is beneficial to agriculture and pastures, and *M. muntjak* and *T. kanchil* are of value for economic and scientific research. We verified the algorithm’s ability to generalize to daily life scenarios on the public dataset and then extended the algorithm to complex field scenarios.

#### 2.1.2. Dataset Annotation and Augmentation

The SWG dataset provides data annotated as a JSON format file conforming to the requirements of the COCO dataset [40], where detailed information about each image is documented, including the image ID, resolution, category, ground truth box, and other data that are useful for the detection models. The description of the ground truth box is [x,y,w,h], wherein x and y are the coordinates of the upper left corner of the box, and w and h denote the width and height of the box, respectively, which are taken as the coordinates and lengths of the pixel points of the images in the dataset. Since the image resolution of the SWG7 wildlife dataset varies from 2560 × 1920 to 4992 × 2808, all images will be scaled to the standard size of 640 × 640 during network training. It is also necessary to complete the normalization of the ground truth box data before training. The description of the ground truth box is converted from the original [top-left corner coordinates, width, height] to [centroid coordinates, width, height], and is finally saved in a text file. Assuming that the JSON file provides W and H to indicate the width and height of the image size, then 1./W, 1./H are the width and height of the scaling factor; the width and height of the ground truth box are then multiplied by the corresponding scaling factor, which is the normalized data, and the centroid coordinates of the ground truth box can be calculated by the upper-left corner coordinates and the width and height; then, finally, the same can be done for the normalization of the scaling factor. The advantage of the normalization process is that after the resize operation of the image, it is not necessary to recalculate the pixel coordinates and length data of the ground truth box. In addition, saving the processed annotation data as a text file with one-to-one correspondence with the image simplifies the dataset loading process to a certain extent.

Synthetic data can be used to perform data augmentation to expand small datasets and increase the diversity and richness of training data. In addition, synthetic datasets can simulate different scenarios and data distributions [41], e.g., different species of animals, different background environments, etc., which helps to evaluate the generalization ability of the model. However, data synthesis needs to follow certain conditions [42], especially for wildlife image datasets, which need to be synthesized reasonably according to the biological habits of the animals, such as the position of the animal foreground placed in the image, the suitability of the foreground to the background, and so on, or else it will also affect the model fitting to a certain extent. Therefore, we expand the dataset using only simple digital image processing algorithms. Since the VOC dataset boasts ample data volume and richness, there is no need to expand it, but the SWG7 dataset is relatively small in data volume. There are a total of 1774 images in SWG7, which are divided by species categories according to the ratio of 8:1:1, resulting in 1423 training set images, 177 validation set images, and 174 test set images, respectively. The dataset is further expanded through horizontal flipping, HSV color space enhancement, and Gaussian noise/blur, as depicted in Figure 3. The method enriches and enhances the information of the images to some extent by expanding the training, validation, and test sets of each object category by three times, respectively. The augmented SWG7 dataset has a total of 7096 images, containing 5692 training set images, 708 validation set images, and 696 test set images, which improves the generalization ability of the model and enhances the detection effect.

### 2.2. Experimental Environment and Parameter Settings

Model training was conducted using an Intel Core i5-10400F 2.90 GHz/12-core CPU (Intel, Santa Clara, CA, USA) with 16 G of RAM, and an NVIDIA GeForce RTX 2060 GPU (NVIDIA, Santa Clara, CA, USA) with 6 G of VRAM, Ubuntu 20.04 operating system. The deep learning framework includes PyTorch 1.12.1, CUDA 11.7, and cuDNN 8.5. The deployment device for the quantized model was chosen to be the NVIDIA Jetson Xavier NX (NVIDIA, Santa Clara, CA, USA), configured with an ARMv8 Processor rev0/6-core CPU with 8 G of RAM, carrying 384 CUDA Cores and 48 Tensor Core-integrated GPU. The Jetson device has multiple power consumption modes that can be switched, and the setting in this paper is 15 watts, 6 cores. The model quantization tool uses TensorRT [43], version 8.4.3.1. Using mosaic enhancement [44] further accelerates the training process and improves the model generalization ability. The input image for the training model is 640 × 640, the number of training epochs is set to 300, and the batch size is 16. Let the initial learning rate be 0.02, which is dynamically adjusted using a cosine annealing strategy with a weight decay of 0.0005 during the training process. The network training is converged using a Stochastic Gradient Descent (SGD) optimizer that incorporates momentum, which has a value of 0.937. In addition, warm-up is also used, grouped by weight decay and Exponential Moving Average (EMA) to optimize the training process.

### 2.3. WildLife Detection Network

#### 2.3.1. Re-Parameterized Convolution

Structural reparameterization [45,46,47] is a representative method for transforming network structures that takes advantage of the additivity feature of the linear computation of convolutions to equate more complex structural transformation parameters with relatively simple structures. Re-parameterized convolution introduces linearly addable branches at training time to integrate more implicit features to enhance representation and improve performance, and multiple network branches also avoid gradient vanishing; at inference time it equivalently reduces the training structure to the same single-branch structure as the original model for inference to reduce network fragmentation without any additional computational or memory overhead. Taking the RepVGG [46] block reparameterization process demonstrated in Figure 4 as an example, to realize structural reparameterization, special multi-branching modules need to be designed with only linear operations inside, which usually include convolutional computation and Batch Normalization (BN), and the nonlinear activation functions need to be placed outside of the multi-branching modules.

The internal computation of the multi-branching block can fuse the convolution and the BN layer, which is also the routine operation in operator fusion. The reparameterization process is first to fuse the BN layers, then fill the 1 × 1 convolution and the identity shortcut shaped as a 3 × 3 square kernel, and finally, since each branch performs linear operations internally, the fusion of the multi-branch weight parameters can be summed up directly. The structural design of the DBB [47] provides paths of varying complexity, which strengthens the feature extraction capability of the 3 × 3 convolution after the reparameterization (Figure 5, upper middle), but the use of the ResNet-like [48] approach to build networks with shortcut connections and a complex branching structure makes the method very time-consuming to train, and we can understand this from the Inception structure of GooLeNet [49]. RepVGG focuses on removing such shortcut connections and uses simple branches to ensure minimal memory access, as shown in Figure 5, upper left. In addition, the asymmetric convolutional [45] “skeleton” helps the network learn stronger spatial information and increase the robustness of the model to rotational distortions, allowing for better generalization to unseen data.

In order to balance the training cost and the improved feature representation ability of the re-parameterized convolutional operator, the 3 × 3 convolution is enhanced using 1 × 3, 3 × 1, and identity-connected (non-downsampling phase with an equal number of input and output channels) branches to preserve more spatial features, called the RepACID structure (Figure 5, upper right). The process of extracting the texture and edge details of the image by RepVGG, DBB, and RepACID is schematically illustrated at the bottom of Figure 5, and is shown in the form of the color of the shades indicating the convolution kernel weights and the pixels value of the feature maps, from which it can be seen that the DBB and RepACID blocks are theoretically stronger than the RepVGG block for the detail richness of the animals in the image.

#### 2.3.2. YOLOv6

YOLO (You Only Look Once) [25], once introduced, has made a big splash in the field of computer vision with its real-time and notable detection accuracy, and has formed a mature architecture after iterative updating of several versions [27,28,44,50], becoming a representative object detection algorithm preferred by industrial applications and deployments. Recently, YOLOv6 [33,51], YOLOv7 [36], and YOLOv8 have been released one after another, which constantly break through the state-of-the-art detector metrics in terms of detection accuracy and speed, etc. Throughout the methods they use, nearly all of them incorporate re-parameterized convolution, indicating that the reparameterization structure has gradually become one of the indispensable methods for designing object detectors with stronger performance. YOLOv6 differs from other algorithms in the YOLO series in that its network design is all based on RepVGG blocks. Taking the small volume model, which is more suitable for deployment in low-computing-power devices, as an example, YOLOv6 has three versions with the same network structure but different widths and depths. The network depth refers to the total number of layers of the network, and the network width refers to the number of convolution kernels in the convolutional layers, i.e., the number of channels. Figure 6 demonstrates the structure of the Backbone network EfficientRep [52] for the small volume model YOLOv6-N/T/S, which has a concise inference structure after reparameterization. Among these three versions, YOLOv6-N has the lowest number of network parameters and computations. Therefore, for efficient detection, the base version chosen in this paper is YOLOv6-N.

We retain the YOLOv6-N simplicity of the Backbone network design, and use the improved RepACID block to replace the RepVGG block as the feature extraction block of the Backbone network to enhance the feature map characterization capability. Since the feature map of Neck is “slim” [53], with little detail information about the target, and the number of channels is large, which contains high-level semantic information, it is more suitable to use 1 × 1 convolution to integrate the channel information; we retain the identity connection to the RepVGG block, which is also inside the multi-branching block, and the 1 × 1 branch is used to learn the semantic information in the channels. 

#### 2.3.3. Inferring Faster Neck and Head

The Neck of YOLOv6-N is fused with feature maps after 8×, 16×, and 32× downsampling, respectively, corresponding to small, medium, and large objects for multiscale detection, and three anchors are clustered using the K-means algorithm for each scale for the SWG7 dataset, which are as follows: small: (32, 60), (82, 70), and (70, 159); medium: (140, 139), (144, 294), and (245, 202); and large: (278, 381), (448, 237), and (514, 429). The Best Possible Recall (BPR) [54] metric is 0.993, and these anchors are also suitable for training the VOC dataset. Due to the large resolution of the images in the SWG7 dataset and the definition of the COCO dataset for small objects as less than 32 × 32 pixels [40], to further reduce the model inference time, the Head of YOLOv6-N is reduced from 3 to 2, as shown in Figure 6. Removing the Head corresponding to 8× downsampling allows the detector to focus on deeper features with a higher number of channels that contain rich semantic information, and the perception of details is complemented by the RepACID block. Meanwhile, reducing the original number of a priori boxes with 9 different scales to 4, using the K-means algorithm to re-cluster the a priori boxes suitable for SWG7, the results obtained are (55, 51), (84, 89), (140, 71), and (216, 158), with a BPR metric of 0.9953. This not only reduces the number of convolutional layers of the YOLOv6-N and the computational complexity, but also reduces the overall computation and post-processing time of the network during the inference time. On the other hand, YOLOv6 uses the Anchor-Aided Training (AAT) strategy, which pays more attention to the anchor, and re-clustering the appropriate priori boxes helps the model training.

#### 2.3.4. Box Regression Loss Function Improvement

The loss function used by the object detection algorithm can affect the performance of the model to a certain extent, which generally contains the following two sub-tasks: classification and localization, corresponding to two loss functions, classification loss and box regression loss, where the box regression loss provides significant learning signals localizing bounding boxes precisely, and it is mostly based on the continuous improvement of the Intersection over Union (IoU) ratio, which is obtained by dividing the intersection of the predicted box and the ground truth box by the concatenation of the two frames. YOLOv6-N uses the SioU Loss [55] as the box regression loss function, which introduces the angle information of the ground truth and predicted frames, while the Loss computation considers three aspects of angle loss, distance loss, and shape loss. However, SioU Loss does not have consistency in choosing to optimize toward the *x*-axis or *y*-axis with a threshold of π/4. In addition, SioU Loss is computationally large and has several hyper-parameters that need to be manually adjusted, which are not conducive to the training of the network.

To solve the above problems, the training phase utilizes CioU Loss [56] and Distribution Focal Loss (DFL) [57,58] to improve the bounding box regression loss function. Among them, CioU Loss takes the aspect ratio of the box and the diagonal distance of the centroids into account, which are less divergent and less computationally intensive compared to SioU Loss without the need to manually set the hyper-parameters, and are suitable for training the generalization on different datasets because of this. The DFL simplifies the continuous distribution of the detection box position into a discrete probability distribution, which takes the ambiguities and uncertainties in the data into account without introducing any other strong prior, especially in the case of ambiguous ground truth box boundaries, which helps to improve the accuracy of bounding box localization.
(1)$$L_{CIoU}=1-IoU+\frac{\rho^{2}\left(b,b^{gt}\right)}{c^{2}}+\beta v,$$
(2)$$v=\frac{4}{\pi^{2}}\left(arctan\frac{w^{gt}}{h^{gt}}-arctan\frac{w}{h}\right),$$
(3)$$\beta =\frac{v}{\left(1-IoU\right)+v}$$
where b and $b^{gt}$ are the centroids of the prediction and ground truth boxes, respectively, $\rho^{2}$ is the square of the Euclidean distance between the two centroids, $c^{2}$ is the square of the length of the diagonal of the smallest outer rectangular box that contains the two boxes, and $w^{gt}$, $h^{gt}$ and w, h are the width and height of the ground truth and prediction boxes, respectively. In addition, due to the small number of wildlife samples in the SWG7 dataset, an attempt was made to further improve the CioU Loss using α-IoU Loss [59], which is more robust to small datasets and noisy detection frames, by adjusting α to give the detector more flexibility in achieving different levels of bounding box regression accuracy.
(4)$$L_{\alpha -CIoU}=1-IoU^{\alpha }+\frac{\rho^{2\alpha }\left(b,b^{gt}\right)}{c^{2\alpha }}+{\left(\beta v\right)}^{\alpha }$$

The YOLOv6 prediction frame target category error loss was calculated using Varifocal Loss (VFL) [60]. It has achieved the state-of-the-art result in several works, and this classification loss function is still followed in this paper.

#### 2.3.5. Quantization of Re-Parameterized Structures

If the algorithm needs to be deployed on an NVIDIA Jetson device to work with an infrared camera that requires a quantization step for the model, the worst part is that the models constructed using the re-parameterized convolution suffer from the INT 8 quantization problem [61], which makes it difficult to maintain the detection accuracy through Post-training Quantization (PTQ) due to structural transformations that lead to a quantization-unfriendly distribution of convolutional kernel weighting parameters [34]. Quantization-aware Training (QAT) uses the entire training dataset to optimize the network for quantization, constraining the weight parameters of the equivalent convolutional kernel to be in the range of [−128, 127] denoted INT 8 [62]. RepOptimizer [34] trains VGG-like models directly via Gradient Reparameterization (GR) without any structural transformations, so quantifying a model trained with RepOptimizer is as easy as quantifying a regular model without its structure re-parameterized.

Unlike RepVGG using RepOptimizer, the RepACID block we improved removes the 1 × 1 branch and adds two asymmetric branches in the re-parameterized convolutional block, which requires custom changes to the RepOptimizer. We denote the modified re-parametric optimizer as RepACID-Optimizer, replacing a CSLA block with a single operator, achieving equivalent dynamic training by multiplying the gradients by some multipliers derived from constant scales. We also refer to such multipliers as Grad Mult (Figure 7). Modifying the gradients with Grad Mult can be viewed as a concrete implementation of GR.

Let $\alpha_{square}$, $\alpha_{hor}$, and $\alpha_{ver}$ be the constant scalars of the 3 × 3, 1 × 3, and 3 × 1 convolution kernels, respectively, W be the 3 × 3 convolution kernel, X and Y be the inputs and outputs, and * represent the convolution computation. The computational flow of the CSLA block is $Y_{CSLA}=\alpha_{square}\left(X*W^{\left(A\right)}\right)+\alpha_{hor}\left(X*W^{\left(B\right)}\right)+\alpha_{ver}(X*W^{\left(C\right)})$. For the GR approach it is $Y_{GR}=X*W\prime$, where W′ represents the weights obtained by the direct training of the parameterized target structure. In order to ensure that $Y_{GR}^{\left(i\right)}=Y_{CSLA}^{\left(i\right)},\forall i>0$, i represents the number of training iterations, and W′ should be initialized as $W^{’(0)}=\alpha_{square}W^{\left(A)(0\right)}+\alpha_{hor}W^{\left(B)(0\right)}+\alpha_{ver}W^{\left(C)(0\right)}$. In addition, when CSLA uses a regular SGD optimizer to update the parameters, the gradients of the GR should be multiplied by $\left(\alpha_{square}^{2}+\alpha_{hor}^{2}+\alpha_{ver}^{2}\right)$, such that L is the objective function, λ is the learning rate, and the W′ update formula is $W^{’(i+1)}=W^{’(i)}-\lambda (\alpha_{square}^{2}+\alpha_{hor}^{2}+\alpha_{ver}^{2})\frac{\partial L}{\partial W^{’(i)}}$. For the pure VGG-style model trained using a RepACID-Optimizer in this paper, the CSLA structure is implemented by replacing the BN behind 3 × 3, 1 × 3, and 3 × 1 convolution layers with constant channel-wise scaling, and replacing the BN in the identity branch with trainable channel-wise scaling (Figure 7). We give the formula for training a single 3 × 3 convolution corresponding to the Grad Mult of the CSLA module according to RepOptimizer. Let C be the number of channels and $R^{C}$ be the feature maps. $s,h,v\in R^{C}$ represents the constant channel scaling after 3 × 3, 1 × 3, and 3 × 1 layers, respectively, while the Grad Mult is denoted by $M^{C\times C\times 3\times 3}$:(5)$$M_{c,d,p,q}=\left\{\begin{array}{c} 1+s_{c}^{2}+h_{c}^{2}+v_{c}^{2},\;\;\mathrm{if}\;c=d\\ \;\;s_{c}^{2}+h_{c}^{2}+v_{c}^{2},\;\mathrm{if}\;c\neq d\\ \;s_{c}^{2},\;\;\mathrm{elsewise} \end{array}\right.$$

The “skeleton” of the 3 × 3 kernel is associated with the 1 × 3 and 3 × 1 branches, and since the scale of the trainable channel can be viewed as a depth-wise 1 × 1 convolution followed by a fixed scale factor of 1, as in RepOptimizer we perform a plus 1 operation on the Grad Mult “diagonal” position when the output shape matches the input shape. The process of associating the hyper-parameters of the optimizer with the trainable parameters of the auxiliary model and search is called Hyper-Search, and it is constructed by replacing the constants in the CSLA structures corresponding to the RepACID-Optimizer with trainable scales (Figure 7 bottom). After the Hyper-Search, we use the searched constants to construct Grad Mults for each operator (Figure 7 bottom, each 3 × 3 layer in the pure VGG structure trained using the optimizer) and store them in the weights file. During the training of the target model on the target dataset, the RepACID-Optimizer multiplies the Grad Mults by elements to the gradients of the corresponding operator after each completion of the forward computation and backpropagation process.

## 3. Results

### 3.1. Evaluation Criteria

To evaluate the effectiveness of our proposed method for public dataset and wildlife image detection, we utilize mAP_0.5_ and mAP_0.5:0.95_ as the accuracy evaluation metrics of the model in percentage (%). AP represents the average precision of the model, which is a widely used evaluation metric in object detection tasks, considering the balance between accuracy (Precision; P) and regression rate (Recall; R) under different confidence thresholds, with a higher value indicating superior performance. mAP is obtained by averaging the APs of all detection categories, where “m” represents the number of categories. mAP_0.5_ is the area under the Precision-Recall (PR) curve at an IoU threshold of 0.5, providing insight into the detector’s precision, but solely at this IoU threshold. mAP_0.5:0.95_ extends the evaluation range from IoU 0.5 to 0.95 in 0.05 increments, capturing the area under the PR curve at each IoU threshold and averaging them. This comprehensive approach better reflects the performance of the object detector across various IoU thresholds.
(6)$$P=\frac{TP}{TP+FP},$$
(7)$$R=\frac{TP}{TP+FN},$$
(8)$$mAP=\frac{\sum_{i=0}^{m-1}\int_{0}^{1}P(R)dR}{m}$$
where TP denotes the number of correctly recognized images, FP denotes the number of incorrectly recognized images, and FN denotes the number of missed images. Finally, the detection speed evaluation metric uses the number of frames processed per second (FPS) or Latency (ms) for single image. In addition, the model size is compared using the number of parameters (Params) and floating-point operations (FLOPs) in millions (M) and gigabits per second (G), respectively.

### 3.2. Results of Improved Algorithm Experiments

#### 3.2.1. Comparison on Public Dataset

At first, we conduct some experiments on a publicly available VOC dataset to validate the performance enhancement of the improved RepACID block after replacing the re-parameterized convolutional block of the YOLOv6-N Backbone, and the results show that the pure VGG-style single-branch model (denoted as YOLOv6-N*) obtained by this method paired with RepACID-Optimizer training has an advantageous detection accuracy. A comparison with the state-of-the-art YOLO family of algorithms [33,36] in terms of Params, FLOPs, and $mAP_{0.5}$ is reported in Table 2. For the comparison, we choose models YOLOv6-N/T, YOLOv7-T, and YOLOv8-T that are close in volume size and start from scratch with 300 epochs using the VOC dataset without pre-trained weights with the same hyperparameter settings. The performance is further improved with a slight increase in computation; in particular, $\mathrm{mA}P_{0.5}$ is improved by 1.4%. Compared to YOLOv7-T, it has obvious advantages in all metrics. Although the detection accuracy is still 0.4% lower than that of YOLOv8-T, our single-branch model is smaller in volume size and has the advantage of application deployment.

#### 3.2.2. SWG7 Dataset Ablation Study

We designed a series of ablation experiments to validate the performance of the improved algorithm and tested it on the SWG7 dataset with and without the RepACID block, the 2 detection Heads, CIoU Loss, and α-IoU Loss. The baseline model YOLOv6-N uses RepVGG blocks for the Backbone and Neck, three scales for the detection Head, and the bounding box regression loss function uses SIoU Loss by default. All networks were trained from scratch without using pre-trained weights with the same hyper-parameter settings. The training, validation, and testing datasets were obtained from the dataset described in this paper to control the variables and ensure the validity of the results. 

Using the RepACID block of this paper, the 2 detection Head after clipping Neck, CIoU Loss, and α-IoU Loss as four independent variable modules, the controlled variable method was used to investigate the effect of using different methods to improve the YOLOv6-N. The results for all networks are provided in Table 3. 

Table 3 demonstrates that the structural design of 2 detection Heads in this study can scale down the number of model parameters by 0.24 M, and the FLOPs drop by 1.52 G. The inference speed reaches 367.3 FPS when taking a batch size of 16 on the RTX 2060 device, which is about 60 FPS faster than the baseline model. Although there is a 0.6% improvement in $mAP_{0.5}$, due to the lack of a small-scale detection Head, mAP_0.5:0.95_ decreases by 2.1%, indicating that the model is not accurate enough for the localization of the bounding box when the IoU threshold is large. After replacing the RepVGG block of the Backbone with the RepACID block, there is an improvement in the detection accuracy without additional inference cost, and the mAP_0.5_ and mAP_0.5:0.95_ metrics improved by 1.4% and 1.3%, respectively. Additionally, the model inference speed remains consistent. Meanwhile, the model combining RepACID blocks and 2 detection Heads not only scales down the model capacity, but also improves significantly compared to the baseline model. After that, the α-IoU method is used to perform a power operation based on the CIoU loss function by taking α = 3. From the results, the mAP_0.5:0.95_ metrics enhancement reflects the fact that α-IoU performs more stably on the change of IoU thresholds than the SIoU Loss, but it is rather inferior to the latter when the IoU is fixed at 0.5. Finally, CIoU Loss works best, paired with this paper’s improved method on the wildlife dataset compared to the initial network, with mAP_0.5_ and mAP_0.5:0.95_ metrics improving by 3.1% and 3.3%, respectively, and detection speeds stabilizing at around 366 FPS. 

We verified the effect of modifying on the detection results by further observing the changes in the confusion matrix, as shown in Figure 8. The FP of background refers to the probability of mistakenly treating wildlife that were not originally background as background, resulting in missed detection of the corresponding wildlife, while the FN of background refers to the probability of identifying the background as the corresponding wild animals, falsely detecting objects that were not originally present, and causing false detection. It is evident that the improved model reduces the false detection rate of different species of wildlife, and the environmental false and missed detections in complex backgrounds are also optimized to some extent. Comparing it to the YOLOv6-N, we found that the improved algorithm significantly improved the detection accuracy of *P. pardicolor*, *M. caeruleus*, *M. muntjak*, *M. mulatta*, and *P. nemaeus*, and only *T. kanchil* and *V. zibetha* slightly decreased. As can be seen in Figure 8, the reason for the decrease in the number of correct detections of both species is due to some of them being misidentified as *P. pardicolor*. We hypothesize that this is due to their close shape and size, the fact that *V. zibetha* and *P. pardicolor* have similar biological characteristics, and that the model was trained to fit close to these aspects, making it difficult to distinguish them effectively.

We validated the impact of our improved algorithm on the detection accuracy by further comparing the mAP_0.5_ metrics of each category of species, as shown in Figure 9. Comparing the models before and after the improvement, we found that using the method in this paper more obviously improved the detection accuracy of *P. pardicolor*, *M. caeruleus*, *M. muntjak*, *T. kanchil*, *V. zibetha*, *M. mulatta*, and *P. nemaeus*, with only *T. kanchil* showing a slight decrease in detection accuracy. We speculate that this is because *T. kanchil* is active in environments that are similar in color to itself, making it difficult to distinguish the background effectively. Due to the pruning of the feature fusion network, the removal of the 8× downsampling detection head resulted in a weaker detection of small-bodied wildlife by the network. Overall, the series of methodological improvements in this paper optimized the accuracy of the model in distinguishing multiple wildlife species, which can solve the problem of wildlife detection in complex field environments to some extent.

For the NVIDIA Jetson Xavier NX device, we can use TensorRT for inference quantization. The quantization steps are to first use QAT or RepACID-Optimizer training to get full-precision (FP32) weights before quantization, then perform ONNX conversion, and finally use TensorRT tools to get a half-precision (FP16), or INT8 weights. The results show that the RepACID-Optimizer method is more capable of maintaining the stability of detection accuracy (Table 4).

Finally, we compare our improved model with the state-of-the-art YOLO series of detection models (Table 5). It is found that the YOLO series of methods with lightweight and stable detection performance can achieve fast detection of wildlife in the SWG7 dataset. The improved method further enhanced the model performance on this basis. Compared to YOLOv6-N and YOLOv7-T, our improved algorithm demonstrates fewer parameters and FLOPs, resulting in significant improvements in inference speed and detection accuracy. Additionally, the model weight file size of the improved algorithm is only 8.9 MB, which is suitable for deploying applications to work with infrared cameras. Although the improved model does not have an advantage over YOLOv8-N in terms of the number of parameters and FLOPs metrics, the significant model detection performance gap allows one to overlook this shortcoming.

#### 3.2.3. SWG7 Dataset Test Results

Some of the test results are based on images from the test set of SWG7 (Figure 10). We use different models to infer the images and plot the detection boxes on them, randomly selecting some representative samples to show them one by one, by species category, from top to bottom. Figure 10a–g correspond to the species categories in Figure 2. In a group of every four images, the detection results of YOLOv6-N, the improved algorithm, YOLOv7-T, and YOLOv8-N are shown from left to right. As can be seen in Figure 10, all four models have a good ability to handle clear, nighttime, small, medium-sized, and incompletely captured images; however, our model performs better in terms of confidence. Further observation of the figure shows that our improved model has the highest score among the samples presented, except for Chevrotain (*T. kanchil*), and the reason for the poor detection of Chevrotain (*T. kanchil*) is that it is active during dawn and dusk when the light is low, and that its body size is small, which makes it difficult to effectively distinguish it from similarly colored backgrounds. In the images of the sample detection results presented, YOLOv8-N had a similar confidence score to the improved algorithm, and the highest detection score for Macaque (*M. mulatta*) of the four models, but there were false detections, indicating that the robustness still needs to be improved. The improved algorithm in this paper not only improves the overall detection accuracy of the model, but also improves the problem of false and missed detection to some extent.

## 4. Discussion

The current study presents an improved algorithm for automated wildlife species detection that can be deployed to various nature reserve areas for animal surveys without human intervention. On one hand, it reduces the intrusive human impact on the biological environment, and on the other hand, it reduces the time cost of manual screening and labeling. The current model demonstrates its ability to detect the seven categories of wildlife mentioned in this paper, which vary considerably in terms of body texture, shape, size, color, and morphological features. 

We first introduce an asymmetric convolutional branch based on the RepVGG block structure, which enhances the feature characterization ability of the re-parameterized convolutional block for image texture details by taking advantage of the additivity of the convolutional linear computation. The feature fusion network is then pruned, thus reducing the number of detection heads to two. Finally, the Box regression loss function is changed to CIoU Loss. The improved model simultaneously enhances the feature extraction capability of the backbone network and speeds up the inference of the model. Through validation tests on a subset of seven wildlife classes from the SWG dataset, the improved algorithm has a better overall performance compared to the state-of-the-art object detection models in the same series, with a $mAP_{0.5}$ of 88.3%, and a model size of only 8.9 MB, which realizes high-precision and high-efficiency object detection of wildlife images in the field under complex environments, and is a practical object detection algorithm with practical value. In addition, for the purpose of facilitating the deployment of the detection model to the Jetson Xavier NX device, and to achieve better application results with the infrared camera, we also consider the quantization problem of the model. Aiming at the problem that INT8 accuracy has a large impact on the detector performance, the optimizer RepACID-Optimizer, which is more suitable for fusing the asymmetric branch-weighted parameterized convolutional block, is improved and trained to obtain a pure VGG-style structural model, which successfully maintains the quantization stability of the model with less loss of detection performance compared to the use of the PTQ and QAT approaches, and achieves quantization stability at a speed of only 6.15 ms for single image inference.

Moreover, the proposed model can replace the current state-of-the-art detection models in terms of accuracy and robustness in the presence of various detection challenges such as visual similarity and complex backgrounds. We have done some improvement work on the algorithmic model, but there are still some shortcomings. Specifically, there is still room for further improvement of the current algorithm. Considering the high demand for model inference speed in practical applications, we have neither used the self-attention mechanism nor a deeper and wider network to pursue higher detection accuracy. In addition, the pre-trained weights obtained on suitable datasets and cross-validation methods should also be of great help in improving the model performance, which has not been attempted in our work. Although the current work focuses on a fixed number of wildlife species, the current model can be extended to more generalized automated animal species detection for comprehensive and systematic wildlife surveys.

## 5. Conclusions

In this paper, an improved wildlife detection algorithm based on the YOLOv6-N model and re-parameterized convolution is designed. This study can provide a convenient and effective method of preliminary wildlife detection for wildlife monitors, and hopefully will be helpful to more developers and researchers engaged in wildlife classification and detection. We plan to use embedded NVIDIA Jetson devices to assist camera traps in capturing wildlife images to solve the problem of limited camera storage capacity and the need to manually retrieve the data regularly. At the same time, the Jetson device can coarsely process the monitoring images to generate detection logs, which can then be transferred to the cloud over the network, a solution that is less costly in terms of data transfer and more real-time than transferring monitoring images to the cloud for processing. 

Deep learning models deployed on embedded devices usually need to be used in conjunction with other common techniques to meet the needs of specific application scenarios, such as through the acquisition of sensor data and then performing preprocessing and feature extraction on the collected data to reduce the amount of model computation and storage requirements, or combined with the optimization of communication protocols to reduce the overhead and delay of data transmission to improve the system’s response speed and real-time. However, the challenge is that the limited computing resources and storage space of embedded devices and the system’s stability, reliability, and risk resistance under harsh environmental conditions, such as the field, need to be considered.

## Figures and Tables

**Figure 1 animals-14-01152-f001:**
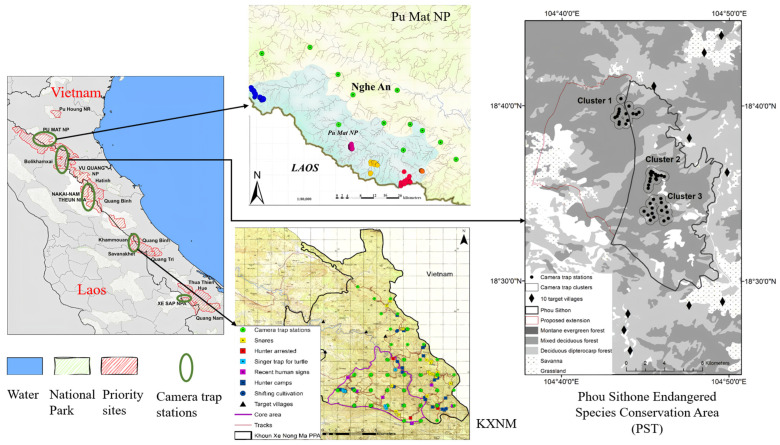
Illustration of camera trap deployment sites.

**Figure 2 animals-14-01152-f002:**
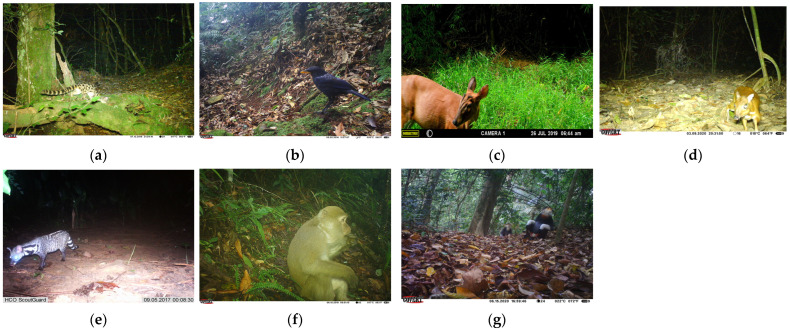
SWG7 wildlife dataset example images: (**a**) *Prionodon pardicolor*; (**b**) *Myophonus caeruleus*; (**c**) *Muntiacus muntjak*; (**d**) *Tragulus kanchil*; (**e**) *Viverra zibetha*; (**f**) *Macaca mulatta*; (**g**) *Pygathrix nemaeus*.

**Figure 3 animals-14-01152-f003:**
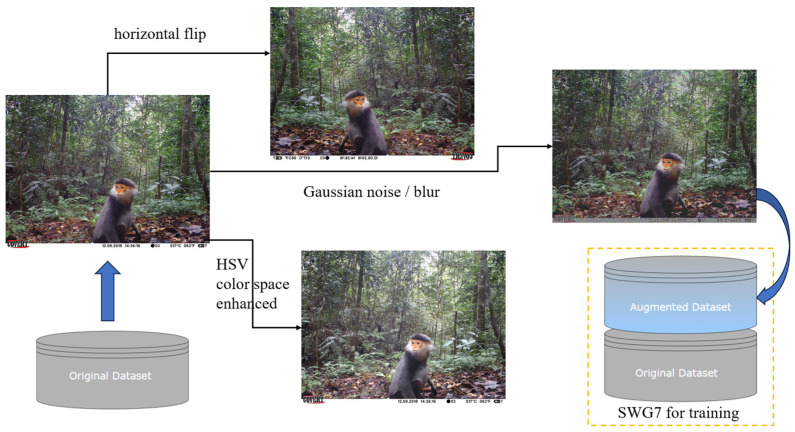
Illustration of dataset expansion.

**Figure 4 animals-14-01152-f004:**
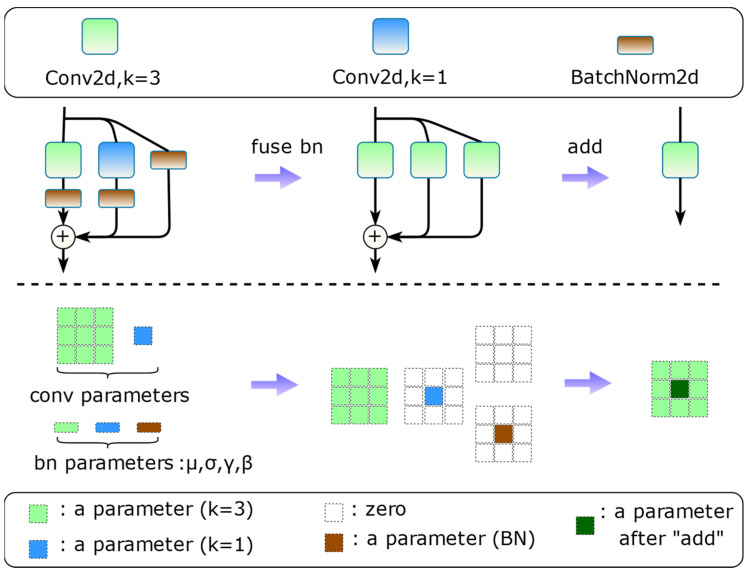
The re-parameterized structure of RepVGG block.

**Figure 5 animals-14-01152-f005:**
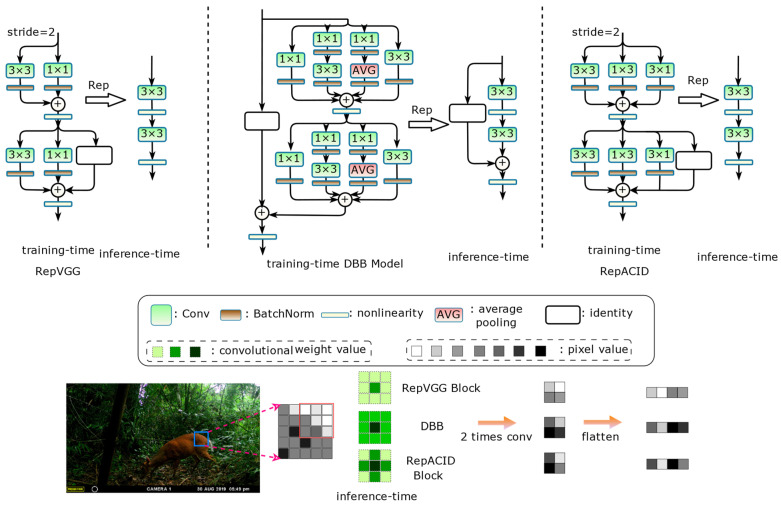
Comparison of the structures of RepVGG, DBB, and RepACID. (The color from light to dark indicates the big or small value).

**Figure 6 animals-14-01152-f006:**
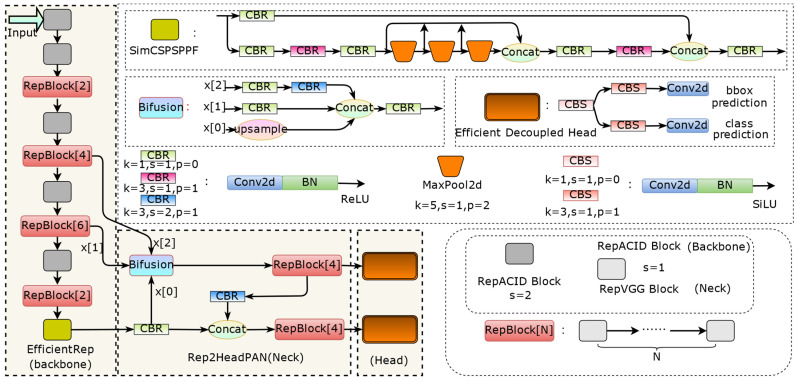
Illustration of the structure of improved algorithm based on the YOLOv6-N model.

**Figure 7 animals-14-01152-f007:**
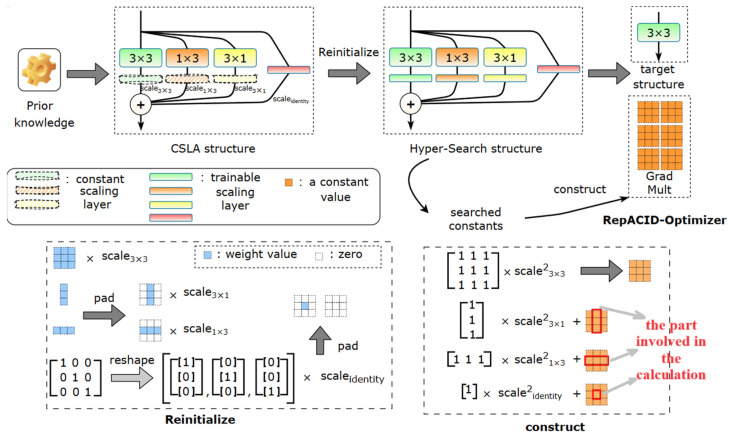
The pipeline of using RepACID-Optimizer. (Scale for the identity branch is set to 1 when the number of input channels is equal to the number of output channels, otherwise, it is set to 0).

**Figure 8 animals-14-01152-f008:**
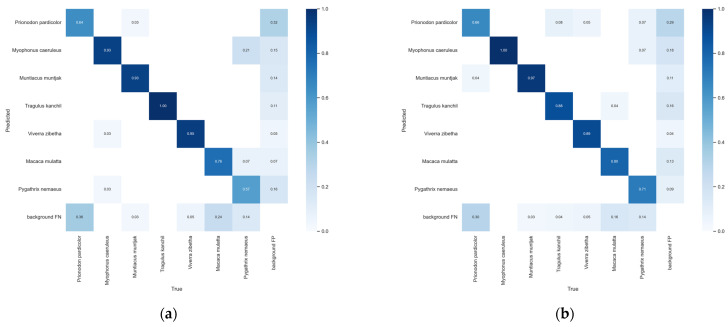
Confusion matrix comparison. (**a**) YOLOv6-N; (**b**) Improved algorithm.

**Figure 9 animals-14-01152-f009:**
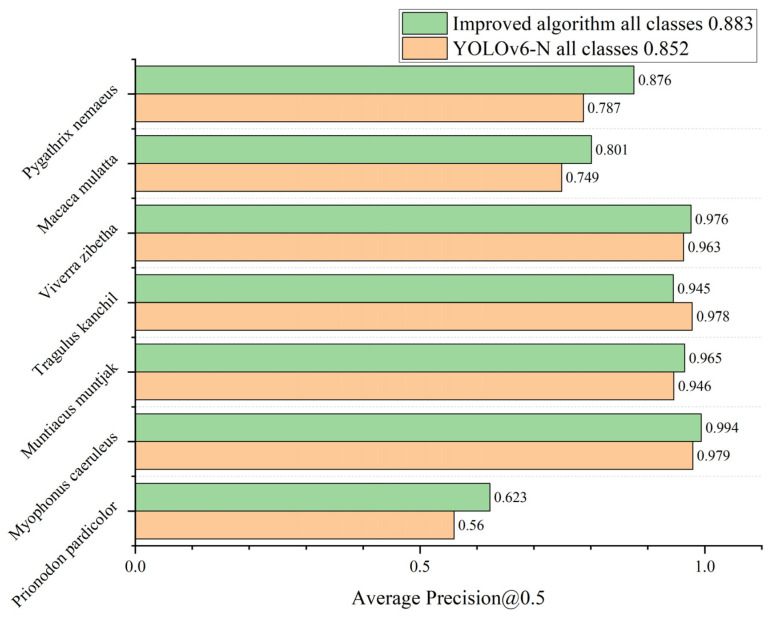
AP_0.5_ of YOLOv6-N and Improved algorithms for all species categories on the SWG7 Dataset.

**Figure 10 animals-14-01152-f010:**
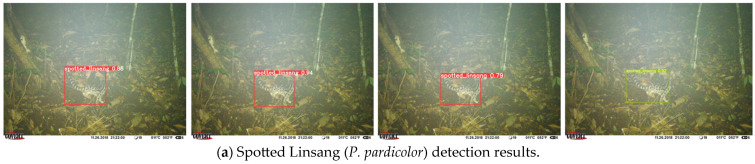
Comparison of combinations on SWG7 test set.

**Table 1 animals-14-01152-t001:** SWG Camera Trap Stations Deployment Information.

Country	Province	Site	Survey Period	Trap Nights	Camera Station
Laos	Khammouane	KXNM	June 2017–November 2019	63,977	311
			February–April 2020	-	270
	Xe Kong	SAP	July 2018–April 2019	26,991	181
	-	NNT	2018	-	29
	Bolikhamxay	PST	June–November 2019	5151	47
		Thongmixay	August–November 2019	497	7
Vietnam	-	Pu Mat	January–December 2018	-	165
			November 2018–December 2019	23,144	152

**Table 2 animals-14-01152-t002:** Performance comparison of several state-of-the-art YOLO family algorithms with the improved model on the VOC validation set. (‘*’ represents the pure VGG-style single-branch model).

Model	Params (M)	FLOPs (G)	mAP_0.5_ (%)
YOLOv6-N	4.63	11.35	80.5
YOLOv6-T	10.42	25.46	84.2
YOLOv7-T	6.06	13.2	80.0
YOLOv8-T	6.42	16.8	82.3
YOLOv6-N *	4.64	11.43	81.9

**Table 3 animals-14-01152-t003:** Results of ablation experiments.

RepACID	2 Detection Head	CIoU	α-IoU	Params (M)	FLOPs (G)	mAP_0.5_ (%)	mAP_0.5:0.95_ (%)	FPS (bs = 16)
-	-	-	-	4.63	11.34	85.2	55.5	306.7
-	√	-	-	4.39	9.82	85.8	53.4	367.3
√	-	-	-	4.63	11.34	86.6	56.8	305.8
√	√	-	-	4.39	9.82	87.3	58.1	365.6
√	√	√	√	4.39	9.82	86.4	58.2	364.3
√	√	√	-	4.39	9.82	88.3	58.8	366.5

‘-’ indicates that this method is not used, and ‘√’ means that it is used. ‘bs = 16’ means the validation batch size on RTX 2060 device is 16.

**Table 4 animals-14-01152-t004:** Improved algorithms use different quantization methods.

Method	Precision	mAP_0.5_ (%)	mAP_0.5:0.95_ (%)	Latency (ms)
PTQ	FP16	87.2	57.1	7.24
	INT8	84.2	55.4	6.21
QAT	FP16	87.5	58.2	7.22
	INT8	85.1	57.5	6.18
RepACID-Optimizer	FP16	87.6	58.2	7.19
	INT8	85.3	57.6	6.15

**Table 5 animals-14-01152-t005:** Performance comparison of different detection methods.

Model	Params (M)	FLOPs (G)	mAP_0.5_ (%)	mAP_0.5:0.95_ (%)	Model Size (MB)	FPS (bs = 16)	FPS (bs = 1)
YOLOv6-N	4.63	11.34	85.2	55.5	10.4	306.7	112.1
YOLOv7-T	6.03	13.2	82.8	50.5	12.3	312.5	131.6
YOLOv8-N	**3.01**	**8.2**	85.5	57.8	**6.2**	294.1	109.9
Ours	4.39	9.82	**88.3**	**58.8**	8.9	**366.5**	**158.7**

‘bs = 16’ and ‘bs = 1’ for RTX 2060 devices. The best results are shown in bold.

## Data Availability

The data used to support the findings of this study are available from the corresponding author upon request. The data are not publicly available due to that they can be easily accessed through the citations in this article, and we hope that more scholars will study this dataset.

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
