# Peer review of "Improved Re-Parameterized Convolution for Wildlife Detection in Neighboring Regions of Southwest China"

_animals, 2024, doi:10.3390/ani14081152_

Round 1

Reviewer 1 Report

Comments and Suggestions for Authors

The topic of the present work is very interesting. It is also in according with the actually technological evolution in the veterinary sector. In fact, the developments of new technologies represents a powerfull tool for a further investigation on animal ecology, behaviour and so on.

The manuscript is well written, anyway some changes have to be performed in order to go ongoing with the publication.

I suggest the Authors to summaryze the abstract. In my opinion is too longer, expecially when they widely describe the methodogy used. It's ok but in any case, you will discuss it in the full text, so don't exceed in explications in the abstract. In particular, from line 27 to  31. 

Introduction. Line 56. I suggest you also to spend few lines about the importance also to monitor wildlife in order to prevent disease spreading and possible problems linked to the wildlife/human interface. These are important keys for a One Health approach. 

Line 132-157: the aim of the work must be better clarified summarising the key points of the research. 

Table 1 could be inserted in supplementary materials.

Line 267. formulas must be reported in a correct format please. 

Results are clear presented, goo work!

Conclusions. This part is well written, anyway I suggest you also to better explain the possible combineted use of this method with ordinary techniques and also if this approach require a good knowledge of the system, what will be the futur challenge in the research system?

Author Response

Response to Reviewer 1 Comments

1. Summary

Thanks very much for the time to review this manuscript. I really appreciate all your comments and suggestions. We have considered these comments carefully and tried our best to address every one of them. Please find the detailed responses below and the corresponding revisions/corrections highlighted/in track changes in the re-submitted files.

Note: All the reviewers’ comments and suggestions are shown in bold, All the removed parts from the original paper are shown in red fonts with a strikethrough, and all the changes and additions are highlighted in yellow.

2. Questions for General Evaluation

Reviewer’s Evaluation

Response and Revisions

Does the introduction provide sufficient background and include all relevant references?

Yes/Can be improved/Must be improved/Not applicable

[It has been improved.]

Are all the cited references relevant to the research?

Yes/Can be improved/Must be improved/Not applicable

Is the research design appropriate?

Yes/Can be improved/Must be improved/Not applicable

Are the methods adequately described?

Yes/Can be improved/Must be improved/Not applicable

Are the results clearly presented?

Yes/Can be improved/Must be improved/Not applicable

Are the conclusions supported by the results?

Yes/Can be improved/Must be improved/Not applicable

3. Point-by-point response to Comments and Suggestions for Authors

Comments 1: I suggest the Authors to summaryze the abstract. In my opinion is too longer, expecially when they widely describe the methodogy used. It's ok but in any case, you will discuss it in the full text, so don't exceed in explications in the abstract. In particular, from line 27 to 31.

Response 1: Thank you for pointing this out. We agree with this comment. Therefore, we have made changes based on your suggestions as follows:

Through an analysis of feature map properties at different stages, We deliberately introduce asymmetric convolutional branches to enhance the feature characterization capability of the Backbone network. Additionally, we streamline the Neck and use CIoU loss to improve detection performance. For quantitative deployment, we refine the RepOptimizer to train a pure VGG-style network. Experimental results demonstrate that our proposed method empowers the model to achieve an 88.3% detection accuracy on the wildlife dataset in this paper. This accuracy is 3.1% higher than YOLOv6-N, and surpasses YOLOv7-T and YOLOv8-N by 5.5% and 2.8%, respectively. The model consistently maintains its detection performance even after quantization to the INT8 precision, achieving an inference speed of only 6.15ms for a single image on the NVIDIA Jetson Xavier NX device.

Comments 2: Introduction. Line 56. I suggest you also to spend few lines about the importance also to monitor wildlife in order to prevent disease spreading and possible problems linked to the wildlife/human interface. These are important keys for a One Health approach.

Response 2: Agree. We have added relevant citations.

In addition, The role of small mammals as carriers or reservoirs of zoonotic diseases in urban forest areas [3]. Therefore, wildlife is monitored also to prevent the spread of disease and possible problems associated with contact with wildlife and humans, these are important for the One Health approach [4].

3. Maydanov, M.; Andreychev, A.; Boyarova, E.; Kuznetsov, V.; Ilykaeva, E. SMALL MAMMALS AS RESERVOIRS OF TULAREMIA AND HFRS IN THE FOREST ZONE OF SARANSK.

4. Mackenzie, J.S.; Jeggo, M. The One Health Approach—Why Is It So Important? TropicalMed 2019, 4, 88, doi:10.3390/tropicalmed4020088.

Comments 3: Line 132-157: the aim of the work must be better clarified summarising the key points of the research.

Response 3: Agree. We modify the last paragraph of the Introduction as follows:

The primary goal of this study is to realize efficient automatic wildlife monitoring, high accuracy, and fast detection and recognition of monitoring images captured by camera traps. We propose a wildlife detection algorithm based on improved YOLOv6 to cope with complex field environments. First, to improve the detection accuracy of the detector, taking the re-parameterized convolution as the research object, we analyze the focus of the detector on the feature map at different stages, use asymmetric convolution to enhance the feature extraction ability of the backbone network’s convolutional block, improve the detector performance with merely an increase in training costs. Second, after pruning the feature fusion network, using the architectural design of the dual detection head to improve the inference speed of the model. Then, the box regression loss function CIoU loss is introduced to enhance the detector accuracy and accelerate the network convergence. Finally, ...

Comments 4: Table 1 could be inserted in supplementary materials.

Response 4: Thanks to your suggestion, we have used Table 1 to display the trap camera information in conjunction with other reviewers' comments.

Comments 5: Line 267. formulas must be reported in a correct format please.

Response 5: We have similarly combined this with other reviewers' comments that "This section has links to published work, so you should not repeat such detailed information, but limit yourself to short descriptions and links. "

Shorten subsection 2.3, Line 267. formulas have been removed.

Comments 6: Conclusions. This part is well written, anyway I suggest you also to better explain the possible combineted use of this method with ordinary techniques and also if this approach require a good knowledge of the system, what will be the future challenge in the research system?

Response 6: We have added the following sentences at the end of Conclusion:

Deep learning models deployed on embedded devices usually need to be used in conjunction with other common techniques to meet the needs of specific application scenarios, such as through the acquisition of sensor data, it's preprocessing, and feature extraction to reduce the amount of model computation and storage requirements, or combined with the optimization of communication protocols, to reduce the overhead and delay of data transmission to improve the system's response speed and real-time. However, the challenge is that the limited computing resources and storage space of embedded devices and the system's stability, reliability, and risk resistance under harsh environmental conditions, such as the field, need to be considered.

4. Response to Comments on the Quality of English Language

We double-checked the manuscript and corrected some word, grammatical, and presentation errors.

5. Additional clarifications

We have also corrected some writing errors and reorganized the order and structure of some subsections of 2.3.

Reviewer 2 Report

Comments and Suggestions for Authors

The authors present a novel method for Wildlife Detection by improving the reparametrization of models. Please find below few comments.

- Can the authors present also results in the form of a confusion matrix?

- What are the potential practical applications and your further plans in that area - improve the final sentence in the conclusions part.

- Did you use any cross-validation in your work?

- Please improve the discussion part - it should be more extended.

- Please comment at least if synthetic data coming from crowd simulations (or any other) could improve the results of your method on the example of DOI: 10.5220/0011692500003417 or any other synthetic dataset, having different animals inside - both in training and validation process.

Comments on the Quality of English Language

not concerned

Author Response

Thanks very much for the time to review this manuscript. I really appreciate all your comments and suggestions. We have considered these comments carefully and tried our best to address every one of them. Please find it in the attached document.

Reviewer 3 Report

Comments and Suggestions for Authors

Dear Authors,

please make changes to the manuscript.

Author Response

Response to Reviewer 3 Comments

1. Summary

We thank you for the critical comments and helpful suggestions. We have taken all these comments and suggestions into account, and have made major corrections in this revised manuscript which we hope meet with approval.

Note: In the revised manuscript, all the responses are shown in blue with bold, all the removed parts from the original paper are shown in red fonts with strikethrough, and all the changes and additions are highlighted in yellow. Please find the detailed responses in the attached document.

2. Questions for General Evaluation

Reviewer’s Evaluation

Response and Revisions

Does the introduction provide sufficient background and include all relevant references?

Yes/Can be improved/Must be improved/Not applicable

[We have made changes based on your suggestions and added relevant citations.]

Are all the cited references relevant to the research?

Yes/Can be improved/Must be improved/Not applicable

Is the research design appropriate?

Yes/Can be improved/Must be improved/Not applicable

Are the methods adequately described?

Yes/Can be improved/Must be improved/Not applicable

[We added information to the wildlife dataset and improved the description.]

Are the results clearly presented?

Yes/Can be improved/Must be improved/Not applicable

[We have made improvements based on your comments.]

Are the conclusions supported by the results?

Yes/Can be improved/Must be improved/Not applicable

[It has been modified.]

3. Point-by-point response to Comments and Suggestions for Authors

Please find the detailed responses in the attached document.

4. Response to Comments on the Quality of English Language

Please find them in the attached document.

5. Additional clarifications

Please find them in the attached document.

Round 2

Reviewer 2 Report

Comments and Suggestions for Authors

Thank you - no further remarks.

Comments on the Quality of English Language

not concerned 

Reviewer 3 Report

Comments and Suggestions for Authors

Dear Authors,

I am satisfied with the corrections made to the manuscript. I accept your answers and corrections positively. Thank you.